# Absenteeism of Healthcare Personnel in the COVID-19 Era: A Systematic Review of the Literature and Implications for the Post-Pandemic Seasons

**DOI:** 10.3390/healthcare11222950

**Published:** 2023-11-12

**Authors:** Helena C. Maltezou, Caterina Ledda, Nikolaos V. Sipsas

**Affiliations:** 1Directorate of Research, Studies and Documentation, National Public Health Organization, 3-5 Agrafon Street, Marousi, 15123 Athens, Greece; 2Occupational Medicine, Department of Clinical and Experimental Medicine, University of Catania, 87 Santa Sofia Street, 95124 Catania, Italy; cledda@unict.it; 3Pathophysiology Department, Medical School, National and Kapodistrian University of Athens, 75 Mikras Asias Street, Goudi, 11527 Athens, Greece; nsipsas@med.uoa.gr

**Keywords:** COVID-19, absenteeism, SARS-CoV-2, healthcare personnel, healthcare workers, vaccination

## Abstract

This systematic review aimed to assess COVID-19-associated absenteeism among healthcare personnel (HCP). PubMed was searched on 4 February 2023. Inclusion criteria were the presentation of original data on COVID-19-associated absenteeism among HCP. Exclusion criteria were absenteeism associated with burnout, mental health illness, post-COVID syndrome, or child-care. Nineteen articles were identified; fifteen concerned almost exclusively the first pandemic year. Hospitals accounted for most data. There was heterogeneity across studies in terms of presentation of absenteeism data. Before COVID-19 vaccines became available, COVID-19 was a major driver of HCP absenteeism with excess costs, while the mean duration of absenteeism ranged from 5.82 to 33 days per episode of absence. Determinant factors of absenteeism rates were department of employment, high-risk exposure, age, profession, and work experience of HCP, suspected COVID-19, SARS-CoV-2 testing, SARS-CoV-2 positivity, and return-to-work strategy. Two studies demonstrated that COVID-19 vaccination significantly reduced the burden of absenteeism. Routine testing of asymptomatic HCP and use of personal protective equipment also significantly ameliorated absenteeism. In conclusion, COVID-19 has been a major driver of HCP absenteeism. Research is needed to assess how COVID-19 will impact HCP in the next years, considering the new SARS-CoV-2 variants, the co-circulation of other respiratory viruses, and the newer COVID-19 vaccines. Networks are needed to survey morbidity and absenteeism among HCP in real-time and guide vaccination policies.

## 1. Introduction

Healthcare personnel (HCP) have been recognized as a high-risk profession for infection as early as 430 BC, as Thucydides mentions in his detailed description of the devastating Plague of Athens [1]. The disproportional risk for infection and fatalities among HCP has been confirmed several times since then, the latest being during the COVID-19 pandemic. According to rather conservative estimates of the World Health Organization, 80,000 to 180,000 HCP died globally between 1 January 2020 and 16 May 2021 [2]. Moreover, on many occasions, HCP contributed to SARS-CoV-2 transmission in healthcare facilities and onset of clusters among patients and employees [3,4]. COVID-19 has been also recognized as a major cause of HCP absenteeism which, in several cases proved crucial, in the context of an unprecedented surge of healthcare demand and hospitalizations, particularly during the first pandemic waves [3,5]. In response to this, various healthcare staffing strategies were investigated [6], telemedicine was widely adopted [7,8], while several countries adjusted their sick-leave policies to address the emerging challenges on healthcare workforce [9]. In this respect, HCP were prioritized for COVID-19 vaccination not only to directly protect them, but also, to indirectly protect their patients, to preserve human resources, and eventually the essential healthcare services [10].

To our knowledge, there is only one published review presenting data on COVID-19-associated absenteeism among other causes of productivity loss of HCP; nonetheless, this review is confined in the first pandemic wave [11]. The objective of this systematic review was to assess the peer-reviewed evidence on COVID-19-associated absenteeism among HCP to support healthcare facilities in the post-pandemic seasons.

## 2. Methods

### 2.1. Search Methodology

This is a systematic review guided by the Preferred Reporting Items for Systematic Reviews and Meta-Analyses (PRISMA) 2020 statement [12]. The PubMed database was searched for articles published as of 4 February 2023 using the following keywords’ combinations: “COVID-19 and absenteeism and healthcare” and “COVID-19 and absenteeism and healthcare personnel”. References of articles identified through this first search round were also reviewed. Inclusion criteria were the presentation of original data on COVID-19-associated absenteeism among HCP (outcome). All studies had to be conducted in healthcare facilities. All study designs (e.g., cohort, case series, cross-sectional) were eligible. Exclusion criteria were absenteeism associated with burnout, mental health illness or post-COVID syndrome, or absenteeism among HCP due to child-care obligations. All eligible studies were included in the analysis to answer the review question.

### 2.2. Data Collection

A standardized form was used for data collection. Extracted items per study were country, study design, study period, healthcare setting, number of employed HCP, number of absent HCP, and rate, duration, and characteristics of COVID-19-associated absenteeism among HCP. No assumptions were made about any missing data.

### 2.3. Review Process

Two authors (HCM and CL) independently screened and reviewed all eligible articles and extracted data. The third author (NVS) reviewed the extracted data and resolved any disagreement to reach consensus. In particular, we read the abstracts of all 141 articles identified through this first round of search and selected 53 articles for full reading, followed by the application of our inclusion/exclusion criteria (Figure 1).

Nineteen articles presenting original data on COVID-19-associated absenteeism among HCP were selected. Data from the 19 eligible articles were collected, analyzed, and grouped as follows: data from the first year of the COVID-19 pandemic; data on the impact of infection control strategies on HCP absenteeism; data on the impact of COVID-19 vaccination on HCP absenteeism; and comparison of COVID-19-associated absenteeism with influenza-associated absenteeism. The mean (or median) duration of absence across the included studies was estimated.

### 2.4. Supportive Data

Information from 40 additional articles on HCP morbidity and/or absenteeism (including influenza-associated absenteeism), COVID-19 pandemic or respiratory syncytial virus (RSV) epidemiology, and data from 5 official websites were also used to support the introduction, results and discussion sections. Data from studies published from the 2009 influenza A(H1N1) pdm09 pandemic onwards were used for comparison with COVID-19-associated absenteeism data. Overall, 64 references were included in the reference list of the present review (Figure 1).

### 2.5. Definitions

Infection was defined as a laboratory-confirmed infection with or without symptoms. COVID-19 was defined as symptomatic SARS-CoV-2 infection. COVID-19-associated absenteeism was defined as absence of an HCP from work due to asymptomatic SARS-CoV-2 infection, COVID-19, or for isolation purposes following exposure to SARS-CoV-2. Influenza-associated absenteeism was defined as absence from work due to influenza infection or following exposure to influenza. HCP were defined as all personnel employed in a healthcare facility regardless of contact with patients or type of employment. Presenteeism was defined as attending work while being ill [13].

## 3. Results

Nineteen articles reporting original data on COVID-19-associated absenteeism among HCP were identified [3,14,15,16,17,18,19,20,21,22,23,24,25,26,27,28,29,30,31]. The characteristics of these articles and their main findings are shown in Table 1.

Fifteen articles concerned almost exclusively the first year of the pandemic before COVID-19 vaccines became available, including eight articles confined to the first pandemic wave. Hospitals accounted for most published data. Overall, there was heterogeneity across studies in terms of presentation of absenteeism data. For instance, the total number of HCP, the number of absent HCP, the duration of absence, and the precise study period (at the level of dates instead of months) were not available in all studies, while in some studies excess absenteeism (compared with previous years) was presented.

### 3.1. Absenteeism in the First Year of the COVID-19 Pandemic

Studies from the first COVID-19 pandemic wave indicated considerable absenteeism rates among HCP [3,14,15,16,18,26,27,28]. A retrospective study based on sickness data from a general hospital in London revealed 39% (128 out of 328) of frontline physicians had at least one episode of sickness absence (138 episodes in total) from 16 March through 26 April 2020, summing to a total of 1240 days of absence [16]. At that time period, the sickness absence rate in this hospital was estimated at 9.1% compared with a mean sickness absence rate of 1.5% in the National Health System of England in January 2020 [16]. This study also revealed great variations in sickness absence rates between departments, ranging from 17.1% in the intensive care unit (ICU) to 49.0% in the general medicine department [16]. Although sick absence included both COVID-19-associated and non-COVID-19-associated sickness absence, the recorded excess absence revealed the surge impact of the first wave on the already stressed healthcare system [16]. A prospective study of a cohort of 3398 HCP exposed to SARS-CoV-2 in their workplace in Greece found 41% of them were absent from work for a mean of 7 days; absenteeism rate reached 74.8% among employees with high-risk exposure, while in some cases sick absence extended up to 60 days [3]. In terms of timing, 90.9% of HCP who developed COVID-19 did so by the end of the first week post-exposure [3]. These findings paved the adoption of a 7-day exclusion policy, instead of 14 days, for HCP with high-risk exposure but also in other working settings where the pressure of the pandemic on human resources might be crucial. A survey in the emergency departments of 246 public hospitals across Spain found more than 5% of physicians, nurses, or other emergency employees were on sick leave: 20%, 19%, and 16% during the first pandemic wave, respectively, particularly in regions with high SARS-CoV-2 activity in the community [14]. In response, the nursing personnel and the physicians increased by 83% and 59% in the emergency departments, respectively [14]. Notably, nursing personnel, physiotherapists, and speech therapists have also been identified as the group at highest risk for SARS-CoV-2 infection and absenteeism in a hospital in Brazil, which was mainly attributed to the close patient contact of these professions, including direct exposure to patients’ airways [31]. Similarly, of 977 HCP employed in primary healthcare centers in Sao Paolo, Brazil, 633 (64.79%) used a medical certificate for absence in 2019 compared with 837 (85.67%) in 2020 (total of 17,404 days of absence in 2019 compared with 36,906 days in 2020) [24]. Overall, respiratory diseases as a cause of absence increased significantly in 2020, which is attributed to the fact that primary HCP were highly exposed during the pandemic [24].

The duration of absenteeism in the healthcare workforce also increased considerably during the COVID-19 pandemic. In particular, during the first pandemic year and before the deployment of COVID-19 vaccines, the mean duration of COVID-19-associated absenteeism among HCP ranged from 5.82 to 33 days per episode of absence (Table 1). In an Irish tertiary-care hospital, 44.5% (203 out of 456) of HCP who developed COVID-19 during the first pandemic wave remained symptomatic for up to 57 days, which rendered them unable to present to work for several weeks [28]. During a severe COVID-19 outbreak that occurred in March–April 2020 in six long-term care facilities (LTCFs) in Spain, 24.6% (65 out of 190) of employees were on sick leave for a mean duration of 19.2 days and 69 new employees were hired [15]. The total expenditures to contain the outbreak were 276,281 Euros per month, including 49,748 Euros due to absenteeism and 47,317 Euros for staff replacement (18% and 17.1% of total costs, respectively) [15]. Similarly, absenteeism was the major driver of all costs for the management of 1332 exposed HCP and 252 SARS-CoV-2-infected HCP during the first pandemic wave in the healthcare system of Greece, accounting for 80% of a total of 1,735,830 Euros [32]. Overall, the costs for exposed HCP far exceeded the costs for infected HCP, given the large number of traced and isolated contacts per source of exposure (e.g., a median of 14 HCP were traced per high-risk exposure) [32]. Likewise, a cross-sectional survey in 25 hospitals in Iran estimated out of 22,000 HCP, 1958 (8.9%) HCP were absent due to COVID-19 during the first pandemic wave for a mean of 16.44 days per employee, a total of 32,209 days of absence, and total absenteeism-related costs of approximately US$ 1.3 million [18]. Although both studies calculated the absenteeism-associated costs using the lost wages approach, there was no comparability between them, given their differences in income and healthcare expenditures [18,32]. For this reason, the findings of these two studies could not be generalized to other healthcare systems. The Iranian study also found a negative association between absenteeism, absenteeism costs, being a physician, and work experience [18]. Similarly, in a teaching hospital in Brazil, younger cleaning staff took more sick leaves due to suspected COVID-19 compared with older staff [30]. In contrast, significantly higher absenteeism rates for HCP older than 40 years were recorded in a hospital in Turkey compared with younger HCP [23].

### 3.2. Impact of Infection Control Strategies on HCP Absenteeism

The impact of intensified routine PCR testing of asymptomatic HCP on absenteeism and SARS-CoV-2 transmission was investigated with real-life data and phylogenetic analyses in a hospital in Brazil from March through August 2020 [26]. Absenteeism rates among HCP soared upon implementation of this strategy; however, 3–4 weeks later there was a significant reduction of HCP with COVID-19, a reversal of the absenteeism trend, and a reduction trend of virus transmission to hospitalized patients, despite the peak of virus transmission in the community [26]. The overall positivity prevalence was 17.3% (74 out of 429 HCP), while the duration of sick absence increased by 473% in 2020 compared with 2019 [26]. Therefore, routine screening of asymptomatic HCP proved to be a key strategy for the prevention of virus transmission [26]. On the other hand, in a tertiary-care hospital in London, delayed SARS-CoV-2 detection instead of diagnosis upon admission of infected patients and HCP absence due to COVID-19 significantly correlated with the weekly incidence of healthcare-associated COVID-19 in patients during the peak of the first pandemic wave [27]. Overall, the reduction of HCP to patients’ ratio may negatively impact the implementation of infection control measures [27]. Nevertheless, the protective impact of personal protective equipment (PPE) on sick absence due to COVID-19 has been well documented [3]. It is noteworthy that in a hospital in Brazil, only 20% of COVID-19 cases among cleaning staff worked in ICUs where COVID-19 patients were admitted, while the remaining 80% worked in semi-critical or administrative areas [30]. This was attributed to the intensified training and proper PPE usage in COVID-19 dedicated hospital sectors, especially after the first months of the pandemic [29,30]. Moreover, a randomized cluster trial conducted in three highly populated office buildings of a company before the COVID-19 pandemic, found that an alcohol-based hand sanitizer combined with hand hygiene education significantly reduced healthcare claims by more than 20% and had a positive impact on employees’ absenteeism [33]. It is worthy to investigate the impact of hand hygiene on HCP absenteeism as well.

A model-based study using real-life data from the first pandemic wave in The Netherlands confirmed the abovementioned findings [34]. From a baseline scenario predicting a maximum HCP absenteeism of 5.4%, a scenario of PPE use in all hospital wards (assuming 90% effectiveness) predicted a maximum absenteeism of 2.3% (57% reduction) [34]. Overall, this model demonstrated PPE use in all wards was the most effective means to reduce transmission of highly transmissible SARS-CoV-2 variants [34]. The evolution of infection control strategies also impacted HCP absenteeism. For instance, on 14 October 2020, the national Portuguese guidelines changed the return-to-work criteria for HCP, from relying only on two negative PCR tests to the adoption of clinical criteria also. Consequently, the mean duration of HCP disease in an oncology hospital was reduced from 33 to 20 days [29]. Lastly, a survey among 1128 new-work HCP in the healthcare sector of Quebec, Canada demonstrated that increased workload during the fourth and fifth pandemic waves and fear of COVID-19 were significantly associated with increased psychological distress and indirectly with increased absenteeism [22]. In this latter study, recognition in the workplace moderated the psychological stress among HCP and indirectly reduced absenteeism [22]. Therefore, HCP sick leave during the COVID-19 pandemic may be multifactorial and should be addressed accordingly.

### 3.3. Vaccination against COVID-19 and Absenteeism of HCP

Two multicenter studies investigated the impact of COVID-19 vaccination on absenteeism among HCP in Greece [19,25]. In the first study, a cohort of 7445 HCP [of whom 4823 (64.8%) had received 1–2 vaccine doses of the Pfizer mRNA vaccine] were prospectively followed from 15 November 2020 through 18 April 2021 [19]. In this study, there were 11.8 COVID-19-associated episodes of absence from work per 100 unvaccinated HCP compared with 4.7 episodes of absence per 100 vaccinated HCP, while the mean duration of episodes of absence was 11.9 days among unvaccinated HCP compared with 6.9 days among vaccinated HCP (*p*-value <0.001 for both comparisons) [19]. Vaccine effectiveness for fully vaccinated HCP was 94.16% [confidence interval (CI): 88.50–98.05%] against COVID-19 and 66.42% (CI: 56.86–74.15%) against absenteeism [19]. In practice, a primary series of mRNA vaccination prevented almost seven out of ten episodes of absence among HCP during a period of high healthcare demand and hospitalizations, but no influenza activity [19]. The second study was conducted from 14 November 2021 through 17 April 2022 (dominance of Delta and Omicron variants) when a mandatory COVID-19 vaccination policy for HCP was in place in Greece [25]. Using the same network of hospitals and following actively a cohort of 7592 HCP (85.6% had received a primary series plus a booster dose, 12.5% were partially vaccinated, 1.9% were not vaccinated because of medical exemptions), it was documented that boosted HCP had 1.6 fewer days of absence compared with partially vaccinated HCP (mean duration of 8.1 versus 9.7 days; *p*-value < 0.001) [25]. Overall, multivariable regression analyses found that boosted HCP had shorter duration of absence compared with partially vaccinated HCP by an odds ratio of 0.56 (95% CI: 0.36–0.87) [25]. These findings indicated that, beyond self-protection, COVID-19 vaccination preserved human resources in healthcare facilities and contained absenteeism costs, and therefore has implications for policymakers in the upcoming years. Moreover, when more than four months had elapsed since the last vaccine dose, HCP were more likely to be absent from work for longer periods (OR: 1.22, 95% CI:1.02–1.46) [25]. These findings should be considered to define the optimal vaccination timing, not only to maximize HCP protection but also to restrict the impact of absenteeism on healthcare systems. Similarly, beyond protecting against severe outcomes, full (booster) COVID-19 vaccination conferred significant protection against prolonged hospitalization and prolonged work absenteeism to patients hospitalized with COVID-19 [35].

Vaccine-associated side events may also impact HCP. An online survey among 2103 HCP vaccinated with two doses of an mRNA-COVID-19 vaccine in two large healthcare systems in Southern California, reported that 579 HCP (27.5%) developed vaccine-associated side effects which disrupted their work duties, while 380 (18.1%) missed work mostly for up to two days [20]. An electronic survey among 8375 HCP across 89 hospitals in Germany found that 23% of COVID-19 vaccinations let to one or more days of work absence [17]. These findings should be considered in mass vaccination campaigns targeting HCP and other essential work settings, as well as in preparedness and response plans for future pandemics.

Lastly, a 1:1 multicenter randomized trial of vaccination with bacillus Calmette-Guerin (BCG) vaccine or placebo among 1511 HCP exposed to COVID-19 patients in the Netherlands found that during a one-year follow-up, BCG vaccination had no effect on absenteeism due to COVID-19 or due to any cause [21].

### 3.4. Comparison with Influenza-Associated Absenteeism among HCP

Before the COVID-19 pandemic, influenza has been a major cause of HCP absenteeism. A Hong Kong study of a mean number of 6880 HCP from 2004 through 2009 found that the overall sickness absence increased by 8.4% during annual influenza epidemics prior to the 2009 influenza A(H1N1) pdm09 pandemic and by 57.7% during the 2009 pandemic [36]. During a severe influenza season (2017–2018), absenteeism increased by 70% (from 4.05 to 6.68 days/employee) from week 42/2017 to week 17/2018 (epidemic period) compared with week 18/2018 to week 41/2018 (non-epidemic period) among 5300 HCP in an Italian hospital [37]. These differences were even more pronounced compared with three moderate influenza seasons (2010–2013), especially among nurses and HCP in contact with patients [37], with a high economic impact on the healthcare system [38]. Overall, influenza vaccinated HCP had less excess absenteeism compared with unvaccinated HCP (1.74 versus 2.71 days/employee in one study) [37,39].

Nevertheless, findings from the first year of the COVID-19 pandemic reported even higher absenteeism rates among HCP with a considerable financial impact. A cross-sectional study at a university hospital in Brazil using health records from September 2014 to December 2020 estimated that the mean sickness absenteeism increased from 2.97% in the pre-pandemic period to 5.10% during the pandemic, while the mean number of sickness days increased by 2.03 times (from 619 to 1259 days per month) and the daily cost by 2.49 times compared with the pre-pandemic period (*p*-values <0.001 for all comparisons) [31]. Overall, the COVID-19-associated absenteeism among HCP far exceeded the influenza-associated absenteeism recorded in pre-pandemic seasons, in terms of rate and duration of absence [31,40]. Likewise, analysis of data from approximately 114,000 individuals in the United States (USA) revealed that, regardless of profession, more employees were on sick absence in April 2020 than in any month since January 1976, when data on sick absence became available for the first time [41].

## 4. Discussion

We systematically reviewed studies reporting original data on COVID-19-associated absenteeism among HCP published as of 4 February 2023. Our review demonstrated COVID-19 has been a major driver of HCP absenteeism in terms of rate and duration of absence, far exceeding the absenteeism recorded during influenza seasons before the COVID-19 pandemic. For instance, during the first pandemic year and before the availability of COVID-19 vaccines, the mean duration of COVID-19-associated absenteeism among HCP ranged from 5.82 to 33 days per episode of absence (Table 1). Variables that influenced the rate and duration of absenteeism of HCP included the department of employment, a history of high-risk exposure, the age, profession, and work experience of the employee, suspected COVID-19, SARS-CoV-2 testing, SARS-CoV-2 positivity, and return-to-work strategy.

### 4.1. Implications for the Upcoming Seasons

In view of the reestablishment of influenza and RSV activity [42,43,44,45,46], it would be of public health importance to implement networks for surveillance of morbidity and absenteeism among HCP. Such surveillance networks could capture increasing absenteeism trends in real-time and inform healthcare facilities to address it through staff redirection and not jeopardize patient safety. Morbidity and absenteeism data could also inform COVID-19 vaccination policymakers. Lastly, baseline (seasonal) data on morbidity and absenteeism of HCP could also be used in models for planning for surge capacity in the event of severe seasons or future pandemics. The feasibility and efficiency of a syndromic influenza-like illness (ILI)-based surveillance method attributed to cold and seasonal influenza has been assessed in a network of healthcare facilities of approximately 9500 HCP in the United Kingdom [47]. A time-series analysis of data from 2008 through 2013 revealed a similar burden and seasonal pattern between ILI-absenteeism and ILI in the community, while hospital-ILI-absenteeism often started and peaked earlier than ILI in the community [47]. Using payroll data from 2009–2019 from Rochester, Minnesota, Challener et al. further showed that ILI activity in the community significantly predicted unscheduled absences among HCP [13]. Moreover, studies indicated no difference in the clinical presentation and duration of symptoms between influenza-PCR-positive and influenza-PCR-negative HCP with ILI [40]. Multiplex PCR could contribute to surveillance and also drive diagnosis, management, and infection control in healthcare facilities [48].

Presenteeism was also frequent among HCP with respiratory illness in order to address increased healthcare demand [13]. Presenteeism poses a risk for virus transmission, compromised patient safety, reduced productivity, and increased healthcare costs [13,49,50,51,52]. In a Canadian network of nine acute-care hospitals, 94.6% of ill HCP reported working while symptomatic at least one day per influenza season [53]. The most commonly reported reasons for working with symptoms among 1180 HCP with ILI in a US tertiary-care healthcare system were staff shortage, inability for a colleague to replace work, not willing to use time-off, and unaffordable absence [51]. Accordingly, adequate sick absence policies and staffing may enable absence among symptomatic HCP [51]. Presenteeism was also frequent among HCP during the COVID-19 pandemic [3,19,20].

### 4.2. Vaccination Policies for HCP

The effectiveness of mandatory influenza vaccination policies against symptomatic absenteeism was assessed during a 12-week period of each of three influenza seasons (2012–2015) among 2304 HCP working in healthcare facilities with mandatory vaccination policies compared with 1759 HCP working in healthcare facilities with no-mandatory vaccination policies [54]. In this study, Frederick et al. found that as influenza vaccine uptake rates among HCP increased, symptomatic absenteeism decreased [54]. An influenza vaccination-or-mask policy for HCP was implemented in British Columbia, Canada in 2012–2013. Two studies investigated the effectiveness of this policy. The first study used payroll data and self-reported vaccination status from the first five years of this mandatory policy to assess absenteeism rates and found that early vaccinees (before December 1) had lower absenteeism rates (OR 0.874, 95% CI: 0.866–0.881) and fewer days of absence (RR 0.907, 95% CI: 0.901–0.912) during the influenza season compared with unvaccinated HCP [55]. In contrast, HCP who were vaccinated from December 1 to March 31 and had to wear a surgical mask until vaccination, noted similar absenteeism rates with unvaccinated HCP [55]. The second study of a cohort of 10,079 HCP (77% vaccinated) confirmed that unvaccinated HCP were more likely to incur sick absence due to all-cause-illness during the 2012–2013 influenza season compared with vaccinated staff (excess of 3.3 sick hours for each unvaccinated employee or 7854 sick hours in total) [56]. Overall, considering the pre-influenza and influenza seasons, vaccinated HCP had 23,473 fewer hours of sick absence, which translates to a net saving of over $1.25 M, while the cost for the introduction of the new policy was minimal [56].

Two prospective studies conducted in 2020–2021 and 2021–2022 demonstrated that COVID-19 vaccination significantly reduced the burden of absenteeism in a large cohort of HCP, and most likely contained costs [19,25]. Currently, COVID-19 vaccination remains a key intervention to confer protection against severe outcomes and mitigate the HCP morbidity and absenteeism from COVID-19. Bearing in mind the benefits of mandatory influenza vaccination policies for HCP in many healthcare facilities the past two decades [54], it is worthy to assess the impact of mandatory COVID-19 vaccination policies on HCP morbidity, absenteeism, and costs as well. Given that vaccine-derived immunity wanes over time, the optimal timing for booster vaccinations should be defined. Lessons from the COVID-19 pandemic can help healthcare systems to anticipate staff shortages and mitigate the impact of HCP absenteeism, in light of the co-circulation of influenza and other respiratory viruses.

### 4.3. Limitations

The present review has the following limitations. First, most studies were conducted in the first year of the pandemic, before COVID-19 vaccines were available and highly transmissible SARS-CoV-2 variants emerged [57]. Second, all but one study was conducted in high οr upper-middle income countries [58]. However, shortages in HCP have already been a huge problem in the health systems of several low-income countries, even before the pandemic [59]. It is possible that the management of HCP during the COVID-19 pandemic varied across healthcare systems due to issued guidelines, staff shortages, and logistics among others. Third, almost all studies were hospital-based. Fourth, there was no standardized presentation of absenteeism data across studies; therefore, a meta-analysis was not considered feasible. Another possible limitation is that information about COVID-19 vaccination refusal, burnout, mental health illness, or post-COVID-19 syndrome as a cause of HCP absenteeism is not clearly mentioned in all reviewed studies, despite their increased prevalence among HCP during the COVID-19 pandemic compared with the pre-pandemic period [60,61,62,63]. Lastly, the impact of school closures on healthcare workforce because of child-care obligations was not considered [64].

## 5. Conclusions

The COVID-19 pandemic had an exceptional impact on healthcare workforce. We reviewed the published studies on COVID-19-associated absenteeism among HCP. COVID-19 has been and continues to be a major driver of HCP absenteeism despite the availability of COVID-19 vaccines, far exceeding the absenteeism recorded during pre-pandemic influenza seasons. Studies indicated that PPE use and routine screening strategies reduced the rates and duration of sick absence. Moreover, two studies showed that COVID-19 vaccination significantly reduced the COVID-19-associated absenteeism, including the duration of absence. Nevertheless, there are gaps in our knowledge about HCP absenteeism in the post-pandemic seasons, considering the co-circulation with influenza and other respiratory viruses and the development of newer COVID-19 vaccines. Given that vaccine-derived immunity wanes over time, the optimal timing for booster vaccinations should be defined. Networks to follow morbidity and absenteeism in real-time are needed to guide policymakers and healthcare facilities. The impact of mandatory COVID-19 vaccination policies for HCP on absenteeism should be assessed. Further studies on the benefits of newer COVID-19 vaccines on HCP absenteeism and its economic impact are imperative in the upcoming years.

## Figures and Tables

**Figure 1 healthcare-11-02950-f001:**
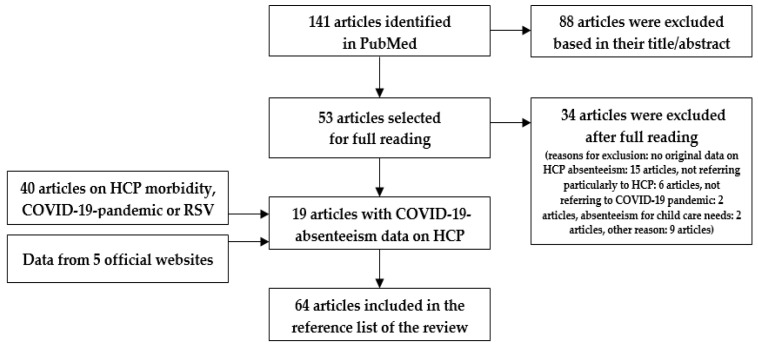
Flow diagram results of literature search.

**Table 1 healthcare-11-02950-t001:** Published articles on COVID-19-associated absenteeism among HCP (as of 4 February 2023).

Country/Study Design	Study Period	Setting	No of HCP	Absent HCP	Duration (Days)	Main Findings
Greece [3]/prospective observational cohort study	13 March–3 May 2020	hospitals, HCCs, laboratories	3398 HCP exposed to SARS-CoV-2	41%; 74.8% post- HR exposure	7 *	HR exposure associated with increased rate and duration of absenteeism
Ireland [28]/case series	first pandemic wave	1 hospital	456 HCP with COVID-19	NS	NS	203 HCP (44.5%) absent for several weeks
Spain [14]/cross-sectional study	March–April 2020	246 EDs of hospitals	NS	>5%	NS	Physicians, nurses, other emergency staff: in sick leave 20%, 19%, and 16% of the time
UK [16]/retrospective cross-sectional survey	16 March–26 April 2020	1 hospital	328 physicians	39%	7 †	Sickness absence rate: 9.1% c/w 1.5% in January 2020; variations between departments
Turkey [23]/cross-sectional survey	11 March 2020–11 March 2021	1 hospital	3967 HCP	24.7%	13.5 †	Absenteeism rate: 1.4%; higher absenteeism: age >40 y.o., HR exposure, SARS-CoV-2 positivity
Brazil [24]/cross-sectional survey	2019–2020	primary HCCs	977 primary HCP	85.67%	9.88 *	64.79% of HCP absent in 2019 c/w 85.67% in 2020; more respiratory diseases in 2020
Portugal [29]/epidemiological survey	11 March–15 December 2020	oncology hospital	2300 HCP	6.7%	33 * before c/w 20 * after 14 October	On 14 October new return-to-work strategy for HCP based on clinical and laboratory criteria
Spain [15]/epidemiological study	6 March–5 April 2020	LTCF	190 employees	24.6%	19.2 *	Severe COVID-19 outbreak: 33.6% attack rate among 198 residents, 69 new employees, 276,281 Euros total cost (35.1% due to absenteeism and staff replacement)
UK [27]/observational study	1 March–18 April 2020	1 hospital	NS	NS	NS	Incidence of HA-COVID-19 correlated with HCP absence due to COVID-19
Brazil [30]/retrospective cohort study	24 March–31 December 2020	1 hospital	NS	199 cleaning staff with 689 medical certificates	5.82 *	44.2% with suspected/confirmed COVID-19; longer absence in suspected COVID-19 c/w other causes (mean 5.82 vs. 3.82 days)
Brazil [31]/cross-sectional survey	September 2014–December 2020	1 hospital	1229 HCP	NS	NS	Mean sickness absenteeism was 5.10% during the COVID-19 pandemic c/w 2.97% pre-pandemic; ×2.03 increased sickness absence duration and ×2.49 increased daily cost during the pandemic
Iran [18]/cross-sectional survey	19 February–21 September 2020	25 hospitals	22,000 HCP	8.9%	16.44 *	Negative association between absenteeism with being a physician and work experience
Canada [22]/cross-sectional survey	18 October 2021–27 January 2022	Quebec healthcare sector	1128 new-work HCP	NS	18.08 *	Increased COVID-19-associated workload and fear of COVID-19 were indirectly associated with higher level of absenteeism
Brazil [26]/time-series analysis	March–August 2020	1 hospital	429 asymptomatic HCP	17.3%	NS	SARS-CoV-2 PCR testing of HCP was associated with reduced absenteeism; duration of absence increased by 473% in 2020 c/w 2019
Greece [19]/prospective observational study	4 January–18 April 2021	5 hospitals	7445 HCP	11.3% among unvaccinated c/w 4.7% among vaccinated HCP	11.9 * in unvaccinated c/w 6.9 * in vaccinated	66.42% vaccine effectiveness of 2 doses of Pfizer mRNA vaccine against absenteeism
Greece [25]/prospective observational study	14 November 2021–17 April 2022	5 hospitals	7592 HCP	28.7%	8.1 * in fully vaccinated c/w 9.7 * in non-fully vaccinated	Fully vaccinated had shorter absenteeism c/w with non-fully vaccinated (OR: 0.56); >17.1 weeks from last vaccine dose associated with longer absence
USA [20]/cross-sectional survey	February–April 2021	2 healthcare Systems	2103 vaccinated HCP	18.1%	93% for 1–2 days	Generalized symptoms, being a nurse and Moderna vaccine recipient were associated with increased risk for work absence
Germany [32]/cross-sectional survey	19 May–21 June 2021	89 hospitals	8375 HCP	23%	NS	Being a male, older age, and BNT162b vaccine recipient were associated with lower risk for work absence
Netherlands [21]/randomized trial	24 March 2020 27 March 2021	9 hospitals	1511 HCP exposed to COVID-19 patients	BCG group: 2.8% placebo: 2.7%	NS	BCG vaccine had no effect on absenteeism

COVID-19: coronavirus disease 2019; HCP: healthcare personnel; Ref: reference; No: number; HCC: healthcare center; SARS-CoV-2: severe acute respiratory syndrome coronavirus 2; NS: not specified; ED: Emergency Department; UK: United Kingdom; c/w: compared with; LTCF: long-term care facility; HA: hospital-associated; vs: versus; OR: odds ratio; PCR: polymerase chain reaction. * Mean duration of absence, † Median duration of absence.

## Data Availability

Data are available upon reasonable request.

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
