# Peer review of "Absenteeism of Healthcare Personnel in the COVID-19 Era: A Systematic Review of the Literature and Implications for the Post-Pandemic Seasons"

_healthcare, 2023, doi:10.3390/healthcare11222950_

Round 1

Reviewer 1 Report

Comments and Suggestions for Authors

The article is adequate both in content and methodology. It is necessary to improve the description of the variables selected in the search for articles, the explanation of their selection and discarding, as well as the tools used to carry out said searches. 

There are 34 articles rejected after a complete reading but the reasons why they have been discarded are not clarified. It would be advisable to expand and improve the diagram in Figure 1.

Author Response

Answer to Reviewer 1

Comments and Suggestions for Authors

Comment 1

The article is adequate both in content and methodology. It is necessary to improve the description of the variables selected in the search for articles, the explanation of their selection and discarding, as well as the tools used to carry out said searches.

Answer to Comment 1

We thank the reviewer for the positive comments and overall efforts to improve our manuscript. As requested, we improved the description of variables selected in the search of articles, and particularly we provided more details about the definition of infection, COVID-19 and healthcare personnel (HCP). We used the PubMed to search for published studies, as reported in Methods (page 2, lines 60-61).     

Comment 2

There are 34 articles rejected after a complete reading but the reasons why they have been discarded are not clarified. It would be advisable to expand and improve the diagram in Figure 1.

Answer to Comment 2

We thank the reviewer for this particular comment. Accordingly, we clarified the reasons for the rejection of the 34 articles in Figure 1.

Reviewer 2 Report

Comments and Suggestions for Authors

this article is well written but there are still several queries need to be addressed first before reviewer could give recommendation:

1. in Line 65, authors mentioned that all study design will be included. Is it for all study design or only for observational study?

2. what is the definition criteria of COVID-19 infection in this paper? 

3. What is the range value of absenteeism  and days found in the included studies?

Author Response

Answer to Reviewer 2

Comment 1

this article is well written but there are still several queries need to be addressed first before reviewer could give recommendation:

  1. in Line 65, authors mentioned that all study design will be included. Is it for all study design or only for observational study?

Answer to Comment 1

We thank the reviewer for the positive comments and overall efforts to improve our manuscript. As requested, we clarify in line 66 (Methods) that all study designs were included in our review.

Comment 2

  1. what is the definition criteria of COVID-19 infection in this paper? 

Answer to Comment 2

We thank the reviewer for this comment. We clarify in Methods the definition criteria of COVID-19 (page 3, section 2.5, lines 100-101).

Comment 3

  1. What is the range value of absenteeism  and days found in the included studies?

Answer to Comment 3 

We thank the reviewer for this comment. Information about the range value of duration of absenteeism is presented in the manuscript (page 7, lines 162-164). Given that the time period of each study and the type of absenteeism recorded (e.g. excess absenteeism) is not consistent throughout the reviewed articles, we prefer not to present data on the range of absenteeism incidence. The issue of not standardized presentation of absenteeism in the reviewed studies is also discussed in the Limitations section (page 11, lines 388-390). 

Reviewer 3 Report

Comments and Suggestions for Authors

Dear authors, 

Thank you very much for the opportunity to contribute to this manuscript. The review focuses on absenteeism in health care providers in the COVID-19 pandemic. 

There are some questions and I would like to add some suggestions to this manuscript.  

1. In the methodology section, the exclusion criteria were burnout and other psychological situations. Concerning the condition that this is a review, how reliable is the information that burnout, depression, and PTBD were excluded? I suggest including this in the discussion section as it may provoke publication errors and bias in the review.

2. The definition of "illness" may be problematic as there are subjective (feeling sick) and objective illness (positive test OR/AND symptoms). I suggest redefining (or discussing) this as it may be a source of error in your work when different authors use different definitions of illness.  (Line 96 and following). In the definition section, the absence of symptoms is problematic as it is highly subjective and prone to bias.     

3. Nations very differently (culturally as legally) treated positive persons in the pandemic. How could that influence your results?

4. Costs are one of the main aspects of your paper. How and on what base were these costs calculated? (Opportunity costs? Are there other "hidden" costs?) Are they comparable concerning data from different countries? Can this be adjusted to raise generalizability? How can these costs be put into context with the extended isolation of patients, too? What about psychological burdens like moral injury, moral distress, and second-victim phenomena in the context of COVID and VACCINATION ("Anti Vaxxers" among staff?) leading to absenteeism (if data is not available, I suggest discussing it)?

I suggest reviewing the whole methods section critically on the calculation of costs and possible bias concerning the reliability of definitions of the different authors. 

5. I suggest clarifying the real role of RSV in this context (p9, L309). 

I think the paper would profit from more clarity in the concise methodology section, especially concerning the definitions.

kind regards

Author Response

Answer to Reviewer 3

Comment 1

There are some questions and I would like to add some suggestions to this manuscript.  

  1. In the methodology section, the exclusion criteria were burnout and other psychological situations. Concerning the condition that this is a review, how reliable is the information that burnout, depression, and PTBD were excluded? I suggest including this in the discussion section as it may provoke publication errors and bias in the review.

Answer to Comment 1

We thank the reviewer for the positive comments and overall efforts to improve our manuscript. As recommended, we added a comment in the Limitations section to address this important issue (page 11, lines 390-393).

Comment 2

  1. The definition of "illness" may be problematic as there are subjective (feeling sick) and objective illness (positive test OR/AND symptoms). I suggest redefining (or discussing) this as it may be a source of error in your work when different authors use different definitions of illness.  (Line 96 and following). In the definition section, the absence of symptoms is problematic as it is highly subjective and prone to bias.     

Answer to Comment 2

We fully agree with the reviewer that there may be bias in relation to the definition of “illness”. As requested, a definition of “illness” was added in section 2.5 (Definitions, page 3, lines 100-101). Nevertheless, our review aimed to record and assess the COVID-19 -associated absenteeism among HCP during the COVID-19 pandemic. COVID-19 absenteeism could be attributed to absence of a healthcare worker due to exposure to a COVID-19 case (for isolation purposes), due to COVID-19 illness or due to asymptomatic SARS-CoV-2 infection. This is clarified in the definitions section (2.5 section, page 3, lines 101-104).      

Comment 3

  1. Nations very differently (culturally as legally) treated positive persons in the pandemic. How could that influence your results?

Answer to Comment 3

We agree with this comment. Indeed, management of HCP during the COVID-19 pandemic may well varied across countries based on issues of guidelines, logistics, and staff shortages among others. To address this issue, we added a comment in Limitations section (page 11, lines 386-388).  

Comment 4

  1. Costs are one of the main aspects of your paper. How and on what base were these costs calculated? (Opportunity costs? Are there other "hidden" costs?) Are they comparable concerning data from different countries? Can this be adjusted to raise generalizability? How can these costs be put into context with the extended isolation of patients, too? What about psychological burdens like moral injury, moral distress, and second-victim phenomena in the context of COVID and VACCINATION ("Anti Vaxxers" among staff?) leading to absenteeism (if data is not available, I suggest discussing it)?

I suggest reviewing the whole methods section critically on the calculation of costs and possible bias concerning the reliability of definitions of the different authors.

Answer to Comment 4

We thank the reviewer for this particular comment. We identified only two studies reporting costs associated with HCP absenteeism during the COVID-19 pandemic. Both studies were conducted during the first pandemic wave and concern costs related to HCP and not to isolation of patients. The first study (Ref 17) was conducted in Greece and the second (Ref 18) in Iran. Given the differences in the healthcare systems and the healthcare costs in these two countries (according to the classification of the World Bank, Greece is an upper income country whereas Iran is a lower-middle-income country), any comparison between the two countries is not feasible. Given that the current review focuses on absenteeism and not on the associated costs, we decided not to present more details in Methods on the estimation od costs in these two studies. In addition, none of these two studies present data on costs related to psychological burden and vaccination (both were conducted before COVID-19 vaccines were available). For this reason, er believe that these findings cannot be generalized to other healthcare systems. Nevertheless, we prefer to present them in order to raise awareness about the expenditures associated with HCP absenteeism during the COVID-19 pandemic.

The issue of psychological burden, long-term COVID-19 and burned out as a cause of absenteeism among HCP is discussed in the Limitations section (page 11, section 4.3). In the limitation section we added a comment on absenteeism related to refusing COVID-19 vaccination among HCP (section 4.3, lines 390-391).        

Comment 5

  1. I suggest clarifying the real role of RSV in this context (p9, L309). 

I think the paper would profit from more clarity in the concise methodology section, especially concerning the definitions.

Answer to Comment 5

A new reference (46) was added in order to discuss RSV epidemiology during and after the COVID-19 pandemic.

As recommended, the definitions were clarified and more details were added in Methods to facilitate the reading of the current review (page 3, section 2.5).  

Round 2

Reviewer 2 Report

Comments and Suggestions for Authors

Authors have answer all of the queries and revise according to reviewer's queries so no the manuscript could be accepted for publication.

Author Response

Thank you so much for your scientific input in order to improve our manuscript.

Reviewer 3 Report

Comments and Suggestions for Authors

Dear authors 

thank you for the opportunity to re-review the article. Most of my recommendations have been addressed. However, I would recommend your response on the economics and comparability of nations for clarification.  

Author Response

Thank you so much for your scientific input in order to improve our manuscript. As recommended, we clarified in the Results section the methodology for absenteeism costs calculation in each study, as well as the issue of comparability between the two countries and the issue of generalizability of these findings to other healthcare systems (page 7, line 102, and page 8, lines 183-186). Please let us know if anything else is needed.